# Factors influencing the early initiation of breast feeding in public primary healthcare facilities in Northeast Nigeria: a mixed-method study

Olukolade George Shobo [ORCID],[1] Nasir Umar,[2] Ahmed Gana,[3] Peter Longtoe,[1] Omokhudu Idogho,[4] Jennifer Anyanti[5]

[1]Monitoring and Evaluation Department, Society for Family Health, Abuja, Nigeria
[2]Disease Control, Faculty of Infectious and Tropical Diseases, London School of Hygiene and Tropical Medicine, London, UK
[3]Office of the Executive Secretary, Gombe State Primary Health Care Development Agency, Gombe, Nigeria
[4]Office of the Managing Director, Society for Family Health, Abuja, Nigeria
[5]Office of the Deputy Managing Director, Society for Family Health, Abuja, Nigeria

**Correspondence to**
Dr Olukolade George Shobo; Shoboolukolade@gmail.com

## ABSTRACT

**Introduction** The early initiation of breast feeding is a high-impact intervention that gives newborns a better chance of survival. We assess the barriers and facilitators influencing the practice of early breast feeding of newborns in public primary healthcare facilities (PHCs) in Northeast Nigeria, to influence the planning of programmes targeted at improving newborn care in the region.

**Method** We used an explanatory mixed-method approach. We conducted case observation of childbirths and newborn care for the quantitative arm, and interviewed mothers and birth attendants 1 hour after childbirth for the qualitative arm. The analysis for the quantitative arm was done with SPSS V.23. For the qualitative arm, we transcribed the audio files, coded the texts and categorised them using thematic analysis.

**Result** We observed 393 and 27 mothers for the quantitative and qualitative arms of the study, respectively. The quantitative arm shows that 39% of mothers did not breastfeed their newborns within 1 hour of birth. The qualitative arm shows that 37% of mothers did not breastfeed within 1 hour of birth. Themes that describe the barriers to early breast feeding in public PHCs are: birth attendants' unwillingness or inability to accommodate mothers' safe traditional practices, ineffective rooming-in practices, staff shortages, lack of privacy in the lying-in ward and poor implementation of visiting-hour policy in public PHCs. The pregnant women denied safe traditional birth practices like chanting, praying or reading religious books during delivery are five times more likely not to breastfeed newborns within the first hour of birth (relative risk=4.5, 95% CI 1.2–17.1) compared with pregnant women allowed these practices.

**Conclusion** Stakeholders must increase their focus on improving breastfeeding practices in public PHCs. Instituting policies that protect mothers' privacy and finding innovative ways to accommodate and promote safe traditional practices in the intrapartum and postpartum period in PHCs will improve the early breast feeding of newborns in these PHCs.

## Strengths and limitations of this study

► We used a mixed-method study design.
► Also, we used purposive sampling technique to select the public PHC, focusing on information-rich cases.
► Data collection included direct observation of mothers' newborn breastfeeding behaviour in the first hour after childbirth.
► We observed a high number of childbirths.
► The sampling approach may limit the generalisability of the findings to places not similar to our study setting.

birth globally.[1] This leaves them vulnerable to diseases and death.[1–3] Newborn deaths continue to account for close to half of all under-5 mortalities across the world.[4] The early initiation of breast feeding, which means breast feeding a newborn within 1 hour of birth,[5] is a high-impact intervention[4 6 7] that gives newborns a better chance of survival.[8] It also provides them long-term health benefits.[9 10] The Northeast region of Nigeria has one of the highest newborn mortalities in the world.[11–13] The early initiation of breast feeding can reduce the risk of these newborn deaths by about a third.[14 15]

The practice of early breast feeding of newborns differs between and within countries.[16 17] For instance, it ranges between 17% and 95% for countries in sub-Saharan Africa.[18 19] In Nigeria, only about 35% of newborns get breast fed within the first hour after birth.[20 21] In the Northeast region of the country, only about 40% of mothers commence the breast feeding of newborns in the first hour after childbirth.[21] In Gombe State, the estimate is 49%.[22] In the rural areas of the country, mothers are more likely not to practice it at all.[20] The mothers' age, level of education and socioeconomic status

## INTRODUCTION

Every year, about 77 million (50%) newborns do not get breast fed in the first hour of

are factors that influence the pattern of early initiation of breast feeding in the general population. Others are maternal and newborn health problems, childbirth method, family support, availability of supplements and maternal preference.[21 23–25]

To improve early breastfeeding and avert newborn deaths, one approach is for more births to occur in health facilities.[14 26–28] Having a birth in a health facility improves the chance that a newborn will breast feed early.[21] Evidence, however, suggests that poor birthing practices occur in health facilities which can disrupt the early start of breast feeding.[17 29–31] While international recommendations for improving breastfeeding practices in health facilities exist,[32] the problems faced by health facilities around the practice are context specific,[33 34] requiring unique responses. Research works focusing on exploring and understanding these specific birthing practices in health facilities are now emerging.[18 29]

The Northeast region of Nigeria is witnessing an increase in health investments by governments and non-governmental organisations[35–38]; aimed at reducing maternal and newborn deaths through improved access to quality childbirth and newborn care services in primary healthcare hospitals (PHCs).[35 36] In Gombe State in Northeast Nigeria for instance, the government is implementing a Village Health Worker programme that improves access to quality obstetric and newborn care services in public PHCs, through community-based demand generation activities. PHC facilities are better positioned to deliver high-impact newborn interventions in Nigeria. They make up 88% of health facilities in the country. In the Northeast, there are 5086 public PHCs, and they make up 87% of the health facilities in the region.[39 40] Understanding the factors that influence the quality and uptake of life-saving newborn care services in these public PHCs is important for improving the effectiveness of the health investments in the region.

In this study, we assess the barriers and facilitators influencing early breast feeding of newborns in public PHCs in Northeast Nigeria. Most studies assessing breastfeeding practice in health facilities have used secondary data from demographic health surveys.[21 24 25 41] While maternal recall is valid and reliable in establishing the period of initiation of breast feeding,[42] the DHS findings do not provide context concerning supply side factors that influenced the maternal behaviour. A recent quantitative study in Bangladesh shows that about 43% of mothers in PHCs in the country do not breast feed their newborns within 1 hour of birth. It did not assess the contextual implementation issues that influence the practice.[31] The literature around the contextual issues affecting the practice in PHCs in Africa is also lacking. Our approach uses a broader study design that helps to develop a deeper understanding of early breastfeeding practices in public PHCs in Northeast Nigeria.[43 44] We believe that our findings will contribute to the discussions about health investments and strategies for improving newborn care in the region, and settings similar to it.

## METHOD

### Study design

We used an explanatory mixed-method approach for this study. Mixed-method study designs deepen how we understand a research phenomenon.[45 46] In the explanatory mixed-method type, a first phase quantitative data collection and analysis is followed by the collection of qualitative data, to explain the quantitative result.[45] The mixed-method approach helps us better understand the factors that influence the early initiation of breast feeding in public PHCs.[45] We conducted the quantitative arm over 4 weeks in December 2017, and the qualitative arm over 1 week in November 2018. Budget constraints delayed implementing the qualitative arm. We assume that the time difference between the study arms is not sufficient to change the practice around breastfeeding newborns in the study setting.

### Study setting

We conducted the study in Gombe State, in the centre of Northeast Nigeria on latitude 9″ 30′ and 12″ 30′N, longitude 8″ 5′and 11″ 45′E. It borders Borno, Yobe, Adamawa, Taraba and Bauchi State. It has 11 local government areas (LGAs) and 114 political wards.[47] There are 603 health facilities across the 11 LGAs in the state, 530 of which are public PHCs.[40] Of the 530 public PHCs in the state, the government has designated 114 as priority PHCs. These 114 are Ward Health Centers and provide basic emergency obstetric and newborn care services. Nurses, community health workers, community health extension workers (CHEWs), junior CHEWs (JCHEWs) and environmental health officers are the main staff of PHCs in Nigeria.[48] Community health officer, a public health nurse, three CHEWs, four nurse/midwives and one medical assistant are the main staff of a Ward Health Center.[49]

Fifty percent of these priority PHCs have staff trained in providing basic emergency obstetric and newborn care services. About 36% of them have labour rooms and lying-in wards. None has a medical doctor, 4% have at least one nurse, and 19% have at least one midwife. Each PHC has an average of six health workers not categorised as skilled attending labour and delivery. These include CHEWs, environmental health technicians, hospital assistants (cleaners) and students. The majority (34%) of the staff in these PHCs have no medical training.[50] Fewer than half of pregnant women in the state access pregnancy care at least four times as recommended, and only about a third access facility-based intrapartum care and/ or skilled attendance at birth.[50] Caesarean sections are not conducted in PHCs in the study setting.

### PHC selection

#### Quantitative arm

We selected 10 of the 114 priority PHCs using a purposive sampling technique. The 10 had the most deliveries per day on the average in the 6-month period prior to starting the study, thus suited to provide rich information on our study objective. They also had labour wards and lying-in

wards. The budget for the study dictated the number of health facilities chosen.

## Qualitative arm

We selected three3 of the 10 priority PHC in the quantitative arm of the study. The three had the most deliveries per day on the average. They also had labor wards and lying-in wards. The intensity of the childbirths in the three PHCs over the study period, will illuminate the barriers and facilitators of early initiation of breast feeding in public PHCs in the region.[51]

### Subject selection

We included all mothers who delivered in the selected PHCs and gave their consent to take part in the study in both arms of the study. For the qualitative arm: we also recruited all consenting healthcare providers who attended deliveries; and the mothers with a live birth for interviews.

### Instrument and data collection

#### Quantitative arm

We recruited and trained nurses and nurse–midwives who are not a part of the health facility staff to complete the assessment tool. They observed and documented the time of events from when a pregnant woman in labour entered the health facility to when she leaves after childbirth or referral. During data collection, they only observed and did not take part or comment in the care offered to the pregnant woman. We only told them to intervene or offer help during a life-threatening situation to the mother and/or baby. The data collectors used separate assessment tools in cases of twin deliveries.

The health facility staff and pregnant women knew of the nurse and nurse–midwives but were unaware of what aspect of care was being observed. Data on the cadre of the health worker, events during the first-to-third stage of labour, and newborn care activities in the first hour after birth were collected. The nurses and nurse–midwives were available for 24 hours in each of the health facilities throughout the study period on a shift schedule. They observed all deliveries.

#### Qualitative arm

We also recruited and trained female nurses and midwives who are not part of the health facility staff to observe and complete the assessment tool under this arm of the study. Observation for each pregnant woman started during the second stage of labour and ended 1 hour after childbirth under this arm of the study. The data collector then interviewed the mother and attendant health worker, after the observation period. The assessment tool used for the observation is an extract from the tool used for the quantitative arm. For the mothers and birth attendants, we asked questions around how they felt about the childbirth process and knowledge of when newborn babies should breastfeed during the interview. We also asked the mothers why they chose to or not to breastfeed their newborns within the hour depending on if they did

or not. Also, we asked the attending healthcare workers why they think breastfeeding occurred or did not occur within the first hour for each newborn. We also asked the attending healthcare workers about the strategies they used to encourage mothers to breast feed their newborns within the hour. The data collectors attended every birth in the health facilities and interviewed respondents who could not speak or understand English in Hausa. They interviewed the mother and attending healthcare worker separately. The mothers' interviews occurred at their bedside in the lying-in ward. The data collectors excused relatives or visitors in the lying-in room during interviews.

### Analysis

#### Quantitative arm

We analysed the data using SPSS V.23. We present nominal variables as percentages. We also determined associations and relative risks (RRs) between initiation of breast feeding and predictor categorical variables using two-by-two contingency tables. To assess significant associations, we used Fisher's or Pearson's $\chi^2$ test as appropriate.

#### Qualitative arm

We transcribed the interviews from their audio files and analysed the data with Saturate, an online qualitative software. Two people on the team analysed a subset of the data and generated codes. The two people then came together to review and agree on the codes generated and their meaning, eliminating less useful codes. The two reviewers held regular face-to-face meetings to discuss their codes. We then analysed the rest of the dataset using the agreed code-framework developed by the two. We generated themes from the codes using an inductive approach.

### Patient and public involvement statement

Resources for patient and public involvement statement were unavailable, so we could not involve patients. The development and dissemination of a policy brief of the study findings will involve patients.

## RESULT

### Respondents' profile

All pregnant women recruited under the two arms of the study consented to be observed. Under the quantitative arm, we observed 393 pregnant women. Most (54%) were between 15 and 24 years old with a median age of 23 years. A quarter were below the age of 20, half below the age of 23 and three quarters below the age of 30. The childbirths were through spontaneous vaginal delivery (SVD). Twin delivery occurred in only six (1.5%) cases. Also, 39% (95% CI 34% to 44%) of the new mothers did not breast feed their newborns in the first hour after delivery. Thirty-three health workers attended the deliveries under this arm of the study. The number of deliveries attended by each health worker ranged from 1 to 37, with an average of 12 (SD 10) deliveries each and a

**Table 1** Pregnant women's profile

| | Quantitative arm % n=393 | Qualitative arms % n=27 |
|---|---|---|
| Age | | |
| <15 | 0 | 3 |
| 15–24 | 53 | 63 |
| 25–34 | 36 | 30 |
| 35+ | 11 | 4 |
| Gestational age (GA) in weeks | | |
| Mean | 41 (SD=12) | 38 (SD=0.6) |
| Mode | 38 | 38 |
| Median | 38 | 38 |
| Parity | | |
| Nulliparous (first pregnancy above 28-week GA) | 20 | 33 |
| Multiparous (more than first pregnancy above 28-week GA) | 80 | 67 |
| Initiated breast feeding within the first hour after birth | | |
| No | 39 | 37 |
| Yes | 61 | 63 |
| Health worker who attended pregnant woman's labour and delivery | | |
| Nurse/midwife | 4 | 0 |
| Junior community health extension worker | 36 | 28 |
| Community health extension worker | 18 | 15 |
| Environmental health assistant/technician/officer | 28 | 19 |
| Hospital assistant | 10 | 19 |
| Nutritionists/dieticians/students | 4 | 19 |
| Sex of health worker that attended pregnant woman's labour and delivery | | |
| Male | 1 | 0 |
| Female | 99 | 100 |

median of eight during the observation period. At least two health workers attended about 61% of the 393 deliveries. Also, JCHEWs attended the majority (36%) of the 393 deliveries (table 1).

Under the qualitative arm, 27 pregnant women were delivered of their babies by 16 health workers. The pregnant women consented to be observed and interviewed. The 16 health workers also consented to be interviewed after the observation. Most (63%) of the pregnant women were between 15 and 24 years old, with a median age of 22 years. A quarter were below 20 years of age and a quarter above 30 years of age. None had a twin delivery. Also, 37% (95% CI 19% to 56%) of the mothers did not breast feed the newborn within 1 hour of birth under this arm of the study. We also interviewed all 27 mothers and 16 health workers 1 hour after childbirth. Of the 16 healthcare workers interviewed under the qualitative arm, most (63%) were hospital assistants. About a third (33%) were CHEWS, 17% (2) were students, 6% (1) were nutritionist and 6% (1) environmental health technician. Of the 27 deliveries under this arm of the study, in 44% of cases, the attending healthcare worker was assisted by another health worker (table 1).

### Knowledge of time to initiate breastfeeding
The qualitative arm shows that health workers know when mothers should breastfeed newborns. Ninety-two

**Table 2** Health workers' response to when breastfeeding should start

| Attending healthcare workers interviewed under qualitative arm | Frequency | Per cent (n=16) |
|---|---|---|
| *Response to when breast feeding should start* | | |
| Immediately after delivery | 9 | 54 |
| 10–30 min after delivery | 2 | 13 |
| 30–60 min after delivery | 1 | 8 |
| 0–60 min after delivery | 3 | 17 |
| 0–24 hours after delivery | 1 | 8 |

percent of them responded that breastfeeding should start between 0 to 60 min (table 2).

They attributed their knowledge to on-the-job training sessions on newborn care with visiting clinical mentors. They believe their knowledge is sustained via peer-to-peer discussions while on the job and during staff meetings. One JCHEW said:

> Breastfeeding should commence immediately after birth …we have clinical mentors that visit and remind us of these things. We also have staff meetings where we remind ourselves of these practices (birth attendant 1)

The word 'immediately' is more used by lower cadre health workers to describe when breastfeeding should start. The CHEWs are more specific about when breast feeding should start. They also link the time breast feeding should start with its benefits. The quote below is a typical response from a senior CHEW when she was asked when breastfeeding should start

> It is very important because it helps the child to suck the yellowish nutrient in the breast milk. It boosts the child's immunity. It also helps the mother's uterus to shrink and close …helping to stop bleeding. It should start by 30 min to 1 hour after delivery (birth attendant 2).

### Barriers to early breastfeeding
#### Birth attendants' unwillingness or inability to accommodate mothers' safe traditional practices
The quantitative study shows that pregnant women denied safe traditional birth practices such as praying or reading religious texts during the second and third stages of labour are five times more likely not to breastfeed within the first hour (RR=4.5, 95% CI 1.2 to 17.1) compared with pregnant women allowed these practices (table 3).

When mother's state of health after childbirth is not an issue, a typical response given by some mothers for not breast feeding in the first hour was the need to first wash the breast or have a bath to feel clean.

> He (the baby) has to exercise patience until we get home (before he is breastfed). I can't breastfeed him before I take my bath (mother 1)

The quantitative study suggests that the mother's need to be clean may not be influenced by the cleanliness of the environment. Initiating breast feeding is not related to the cleanliness and comfort provided by the delivery room. Mothers who had their babies when the labour room was clean and comfortable were just as likely not to initiate breast feeding early as those who had their babies when the labour room was not (RR=0.98; 95% CI 0.7 to 1.4; p=0.9).

#### Poor management of mothers' postdelivery state of health
Postdelivery pains and fatigue are barriers to breast feeding within the first hour after birth. Even when

mothers show a good knowledge of when breast feeding should start, some still express the need to regain strength and wellness first before they breast feed the newborn

> breastfeeding should commence immediately after birth… I did not commence it because I was feeling after pains. The health worker said she will bring the baby to suck. I told her to allow me to have some relief ZA (mother 2)

The quantitative study shows that pregnant women not encouraged to consume fluids or food at least once during labour are twice as likely not to breastfeed within the first hour compared with those encouraged to do so (RR=2.1; 95% CI 1.5 to 3; p=0.001). There is no evidence from the study that blood loss greater than 500 mL during labour and delivery influence early breastfeeding (RR=1.17; 95% CI 0.2 to 5.9; p=1).

#### Human resource shortages
The qualitative study shows that shortage of health workers in PHCs introduces a delay in carrying out newborn care activities. Sometimes, health workers have to attend to other ill patients when there is no one else to assist. When we asked some health workers why a mother under their care did not breastfeed early, a typical response given was:

> You see, if you have someone that will assist you, you will assign the person to carry the baby to the mother and initiate the breast feeding, or weigh the child, or apply chlorhexidine to the baby's cord or any other thing needed while you continue with the remaining work and management of others …but most of the time you are on duty alone. It is because we have shortage of manpower here in this facility (birth attendant 3)

The human resource shortage also affects rooming in in the PHCs. Placing the mother and newborn in the same room after delivery is rooming-in.[52] Sometimes, the health worker has to clean and make this room ready for the mother and newborn. When there is a shortage of staff, this delays the transfer of the mother and newborn to the rooming-in room, and affects early breastfeeding

> I wanted to transfer the mother and baby to the postnatal ward… before she commences breastfeeding (of the newborn). The (lying-in) room is not set (for use yet). I have to clean and make the room so she is comfortable to commence breastfeeding (birth attendant 4)

#### Ineffective rooming-in practices
When rooming-in happens, the quantitative study reveals that mothers who do not have skin-to skin contact with their newborns in the first hour after birth are twice as likely not to breastfeed early, compared with mothers who did (RR=2.3, 95% CI 1.8 to 2.8; p<0.001). Just keeping the mother and newborn in the same room (rooming-in)

**Table 3**  Association between early breastfeeding and predictor variables in the study

| Variable | Response | Women who did not initiate breast feeding within 1 hour after birth | | |
| --- | --- | --- | --- | --- |
| | | % n=154 | Relative risk (95% CI) | P value |
| Birth attendant had received training on newborn care | No | 49 | 1.2 (0.9 to 1.5) | 0.2 |
| | Yes | 51 | | |
| There were delays in providing care | No | 90 | 0.7 (0.5 to 1.03) | 0.2 |
| | Yes | 10 | | |
| Communication was easy and frequent between woman and birth attendant | No | 2 | 0.7 (0.3 to 1.8) | 0.5 |
| | Yes | 98 | | |
| Poor staff attitude | No | 96 | 0.7 (0.4 to 1.1) | 0.2 |
| | Yes | 4 | | |
| Woman denied some safe traditional childbirth practices* | No | 99 | 4.5 (1.2 to 17.1) | 0.003‡ |
| | Yes | 1 | | |
| Birth attendant determined the birth position | No | 41 | 1 (0.8 to 1.3) | >0.9 |
| | Yes | 59 | | |
| Woman was allowed to give birth in the position she preferred† | No | 37 | 0.9 (0.7 to 1.2) | 0.5 |
| | Yes | 63 | | |
| Woman encouraged to consume fluids/ food at least once during labour | No | 16 | 2.1 (1.5 to 3.0) | 0.001 |
| | Yes | 84 | | |
| Mother and newborn kept in the same room after delivery (rooming in) | No | 2 | 2.6 (2.3 to 2.9) | 0.059‡ |
| | Yes | 98 | | |
| Mother had skin-to-skin contact with newborn in the first hour after birth | No | 45 | 2.3 (1.8 to 2.8) | <0.001 |
| | Yes | 55 | | |
| Woman had blood loss greater than 500 mL during labour and delivery | No | 99 | 1.17 (0.2 to 5.9) | >0.9‡ |
| | Yes | 1 | | |
| The labour room was clean and comfortable | No | 18 | 0.98 (0.7 to 1.4) | 0.9 |
| | Yes | 82 | | |

*n=154.
†n=150.
‡Fishers $X^2$.

without skin-to-skin contact has no influence on early breastfeeding (RR=2.6; 95% CI 2.3 to 2.9; p=0.059).

### Lack of privacy and proper visiting-hour policy in the PHCs
The PHCs have open rooming-in rooms that does not guarantee privacy. There are also no defined visiting hours in the PHCs. The qualitative arm of the study shows that male and female relatives visit the new mother in the hospital after childbirth. During the visit, the relatives pray for the newborn baby and congratulate the mother. Some relatives sit around after prayers for long. When relatives come visiting, the mothers have to dress up to receive them and do not breast feed during this time. The birth attendants also delay supporting the mother to breast feed. One birth attendant suggested that addressing the issue puts them in bad light in the community.

  you know, the people in the community have a unique character or attitude. The moment you try to

talk to them about this kind of issue they feel you are molesting them or depriving them of coming close to their relatives. They do not know you are trying to ensure their relative (mother and baby) gets what is beneficial to them (birth attendant 5).

### Facilitators of early breastfeeding
### Health education during ANC and postdelivery period
From the qualitative study, we find that antenatal clinics (ANC) helps pregnant women to learn about breastfeeding newborns within 1 hour of birth. The knowledge they gain during these clinics influences their behaviour after childbirth. Most mothers who practised early breastfeeding said what they learnt from ANC influenced their decision to do so. The typical response they gave is:

  I used to give my children water (after childbirth). I did not know the importance of breastfeeding early.

I used to think breast feeding could start at any time of the day (of birth). I started breastfeeding early because the health workers tell us (of the importance of starting breastfeeding immediately after birth) during ANC (mother 3)

## Encouraging and supporting mothers to start breastfeeding after childbirth

From the qualitative study, we find that when the birth attendants encourage some reluctant mothers to breast feed their newborns within 1 hour of childbirth, they do.

I was told (by the health worker) to give (the baby breast to suck early enough). Normally, I won't (mother 4)

The encouragement process takes the form of a negotiation between the birth attendant and the mother. We find that the information passed by the birth attendants to the mothers around the benefits of early breast feeding facilitate early breast feeding in the PHCs.

I commenced breastfeeding early because of what the health worker said …she explained how it is important to the health of my baby. That's why I commenced it (mother 5)

Sometimes, showing the mothers how to place and breast feed the newborns also facilitates early breast feeding after childbirth in these health facilities

I gently encourage them and tell them to give (breast milk) early… I put the baby on her laps and remove the breast, and demonstrate to her how to breastfeed (birth attendant 6)

## DISCUSSION

In our study setting, we find that close to 4 out of every 10 newborns do not get breastfed within the first hour of birth. This doubles their risk of dying in the first 28 days of their lives.[10 15] Our estimate is four percentage points lower than what researcher observed in PHCs in Asia.[31] It is also higher than estimates among mothers who had SVD in some secondary health facilities, and lower than estimates among mothers who had SVD in some tertiary health facilities, even in Nigeria.[31 53 54] The mixed results emphasise the influence of context on the early breast feeding of newborns in different health facilities. It underscores the need for unique interventions to address the problem.

The northeast region of Nigeria has a shortage of skilled health workers. The Boko-Haram insurgency has made this worse.[55–57] It is also worsened by staff absenteeism in PHCs in the region. Only about 35% of employed staff in PHCs in Gombe are likely to be at work on any given day, for example.[22] Studies suggest that a shortage of human resource and a dominant population of unskilled healthcare workers affects the quality of newborn care in health facilities.[17 58–60] This is not overall consistent with our study findings. We find that unskilled health workers are dominant in our study setting. Also, we find that human resource shortage is a barrier to early initiation of breast feeding in the PHCs. We did not find that the skills of the birth attendants influenced the early breast feeding of newborns. This may be because clinical mentors have trained the birth attendants in our study on newborn care. Educational interventions around support for the breast feeding of newborns have been found to improve health workers' knowledge, attitude and compliance with the practice.[61 62]

In our study, we find that the mothers denied safe traditional birth practices like praying, reciting religious texts or reading religious books during deliveries were five times more likely not to breastfeed the newborns than the mothers not denied. This supports the findings from other settings that shows that the unwillingness of birth attendants to accommodate safe traditional birth practices affects mothers' adoption of supportive care.[59 63 64] This may be because mothers perceive denial of such traditional practices as mistreatment or abuse.[59 65] A study in Norway finds that the recent abuse of women by 'both known and unknown' perpetrators affects their breast-feeding behaviour.[66] Although in our study, we found no evidence to support this. Instead, we find that birth attendants' attitude, rapport skills and negligence during the delivery period do not affect early breastfeeding practice among mothers. The discrepancy in findings may be because the women in our study setting perceive or tolerate abuse differently than women in other settings.[67]

Washing of the breast with water after childbirth is a hygiene-related practised in parts of the world.[68] In our study, mothers express a strong need to wash the breast and/or have a bath before breastfeeding newborns. This makes the 'need for a bath after childbirth' a major theme for why mothers delay breastfeeding newborns in our study setting. Some health workers also agree that mothers should wash their breasts first before breast-feeding newborns after childbirth.[18] Their inability to help the mothers to wash their breasts or have a bath before breast feeding may be because there is no running water in the health facility. Only 38% of priority PHCs in Gombe have running water.[22] The need to wash the breast may be deeper than the need for hygiene, nonetheless. Washing of the breast or having a bath after childbirth before breastfeeding newborns has traditional and/ or religious undertones in other parts of the world.[60 69]

The other barriers that influence early breast feeding of newborn in PHCs that our study finds are post-delivery pains and fatigue within the first hour after birth, delay in skin-to-skin contact during rooming in and poor visiting hour policy in the PHC. These findings reinforce what other studies have reported.[5 21 24 33 41 54 60] The post-delivery fatigue may be because the mother is famished. This may also be due to anaemia, infections, thyroid disorders, mood disorders and cardiomyopathy[70] which our study does not assess. The quantitative arm of our study shows

that mothers not encouraged to take fluids or eat during the second and third stages of labour are more likely not to breast feed early after childbirth. There is no evidence from our study that the fatigue may be due to blood loss. We also find that rooming in is not enough and that skin-to-skin contact must be deliberate. Mothers that did not have skin-to-skin contact with their newborns while in the rooming-in room were more likely not to breastfeed than those who did in our study. Our study re-emphasises that the lack of restrictions on relatives visiting the mothers in the post-delivery period impedes breastfeeding.[71] This could be because the mothers do not feel comfortable breast feeding in front of male relatives or visitors.

Helping mothers to breast feed within an hour of birth is an international recommendation practised by health-care providers in most of the world.[32 72] We find that birth attendants in our study setting practise this and the strategy facilitates the early breast feeding of newborns in public PHCs. This may be because there is an awareness of the recommendation.[73] Our study also reinforces findings from previous studies on the benefits of health education on the timely breast feeding of newborns.[74 75] In our study, we find that health education received by mothers during ANC and the postdelivery period improves the practice of early initiation of breast feeding in public PHCs, in our study setting.

## CONCLUSION

Only about 60% of babies born in public PHCs in Northeast Nigeria get breastfed in the first hour of birth. This means the rest miss important nutrition that saves lives. The stakeholders in the region must increase their focus on improving the breastfeeding practices in public PHCs. This will improve the survival of newborns and impact of their investments. Instituting policies that protect mothers' privacy, and finding innovative ways to accommodate and promote safe traditional practices in the intrapartum and postpartum period in PHCs will improve the early breast feeding of newborns in these PHCs. Birth attendants in PHCs must also be trained on effective rooming-in to further improve early breast feeding of newborns in these public PHCs, however.

**Acknowledgements** The authors thank all the staff of SFH MNCH2 project in Gombe, particularly Magdalene Okolo for their contributions to the study. They also thank the field staff who conducted the observations and interviews. They also thank our partners, the members of the IDEAS team from the London School of Hygiene and Tropical Medicine, for leadership in designing and implementing the quantitative arm of the study.

**Contributors** OGS conceived of the study and developed the original draft of the manuscript and analysed the quantitative data. OGS and PL analysed the qualitative data. NU, AG, JA and OI reviewed, edited and made significant contributions to the development of the final manuscript. All authors read and approved the final manuscript.

**Funding** The study is funded as part of the evaluation of SFH's MNCH2 project being funded by the Bill and Melinda Gates Foundation and implemented in Gombe State. The intervention is funded through donor grants received by SFH. The donor did not play a role in the design of this evaluation.

**Disclaimer** The views and opinions expressed in the paper are those of the authors and do not reflect those of SFH and/or GSPHCDA.

**Competing interests** OGS and PL are consultants working for Society for Family Health (SFH). OI and JA are full-time staff of SFH. AG is the Former Executive Secretary of Gombe State Primary Health Care Development Agency. SFH's programme in Northeast Nigeria seeks to improve MNCH outcomes in the general population.

**Patient consent for publication** Not required.

**Ethics approval** The Gombe State Ministry of Health Ethics Committee granted ethical approval for the qualitative arm of the study (reference no: MOH/ADM/658/VOL.II/104). Mothers and birth attendants also gave their consent to take part in the study before the interviews. The IDEAS team at the London School of Hygiene and Tropical Medicine (LSHTM) got ethical approval for the quantitative arm of the study from LSHTM (reference 6088).

**Provenance and peer review** Not commissioned; externally peer reviewed.

**Data availability statement** Data are available upon reasonable request. Data for the quantitative arm of the study is in a secured database at the London School of Hygiene and Tropical Medicine. Write to Nasir Umar through nasir.umar@lshtm.ac.uk to place a reasonable request for the anonymised version of it. Also, write to Shobo Olukolade, through shoboolukolade@gmail.com to ask for the anonymised transcripts of the qualitative data.

**ORCID iD**
Olukolade George Shobo http://orcid.org/0000-0001-9633-1741

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
