## [Reviewer comments · BMJ Open]

ARTICLE DETAILS

TITLE (PROVISIONAL)	Factors influencing the early initiation of breastfeeding in public primary health care facilities in Northeast Nigeria: A mixed methods study
AUTHORS	Shobo, Olukolade; Umar, Nasir; Gana, Ahmed; Longtoe, Peter; Idogho, Omokhodu; Anyanti, Jennifer

VERSION 1 – REVIEW

REVIEWER	Samwel Maina Gatimu Aga Khan University, Kenya
REVIEW RETURNED	26-Sep-2019

GENERAL COMMENTS	Abstract The authors should revise the introduction and remove the last part "...to influence health investment decisions in the region." The findings may not necessarily influence investment but could influence practice by mothers and health workers. Results: The quantitative findings of the EIBF rate during the qualitative arm of the study should also be reported. Though small, there is a 2 percentage point increase in the EIBF between the two study arms. Use of the term "Lack of privacy..." What do the authors consider as privacy? Are delivering women accorded any privacy? Strength and limitations Page 2, line 33-34: "...the findings do not generalize..." to "...the findings may not..." The last two limitations on generalization could be combined Introduction • Page 3 line 27-28: Revise "...the practice context-specific" to "...the practice are context-specific"• Page 3-4 line 56 and 3: "Recall bias and lack of disaggregation between..... Revise the statement to reflect the correct the available evidence on reliability and validity of maternal recall of breastfeeding. For example, from studies like https://academic.oup.com/nutritionreviews/article-abstract/63/4/103/1940812?redirectedFrom=fulltext. Also, highlight clearly what is limited in the utilization of the study findings from DHS.• Page 4 Line 9-10: "the literature around the practice in PHCs in Africa is lacking." Revise the statement to be clearer and more specific. Practice on what? Method • Clearly, state the actual mixed-method design used in the study• There exists an almost one-year difference between the quantitative and qualitative arms of the study. The author should clearly discuss why the time difference and how it affects the results or interpretation of the results of their study
--

	 • Page 4: Line 42-44: Revise the statement “The picture of maternal and newborn care in the state is comparable to other high mortality settings in sub-Saharan Africa”. The statement is too broad and the comparison with sub-Saharan Africa not right. I suggest the author consider comparing with other Nigerian states or neighbouring countries for a better context. Also, be specific on specific maternal and newborn care indicators. • What is a priority PHC? Provide a detailed description of what priority PHCs are? • What is the average delivery per day in Gombe PHCs and what were the average deliveries per day in those 10 PHCs? This information will help the reader appreciate the purposive sampling of the PHCs as well as the choice of the 3 facilities for qualitative arm of the study. Also, it is good practice to state which facilities are being referred to. This help in replication studies as well as follow-up by interested stakeholders. What was the staffing like in these facilities? • The author should consider providing information that would give the reader a better understanding of management of labour and childbirth in the selected PHCs. A reading of some of the qualitative findings raise a lot of questions on the practices e.g. after delivery, what happens? Is the child taken to a separate room? Is there a birth companion during labour/childbirth? What role do they play if present? What are the roles of the nurses – managing pregnant women and newborns or other patients within the facility?  o How many health care providers assisted in each delivery? o There is need to provide clear conceptual or operational definition of the attendant health worker during labor and delivery e.g. who is a JCHEW, CHEW, Hospital assistant, Environmental health assistant/technician/officer. In addition, provide information about their training and scope of practice e.g. Are they licensed to performed deliveries? What do the Nigeria and Gombe maternal and newborn policies say about them? o Was monitoring of laboring mother done by the same health attendant? What are the documentation practices? • The qualitative data analysis process should be discussed further:  o Page 6 Line 10: Who are two member of the team that performed the analysis? o What informed the choice of grounded theory? Was it the best choice of qualitative approach? o In what language were the interviews conducted? o How was grounded theory used? o At what point was analysis of data performed? o How did the authors ascertain theoretical saturation? o Include either in text or as appendix – the coding process – how the current themes were arrived at. • Page 6. Line 16: Delete the line “To analyze the respondents...” • How was consent obtained without informing the healthcare workers and mother what was being observed? The author should consider including sample information sheet and consent forms used as supplementary materials. • It would be informative for the authors to include the data collection tools as supplementary materials • At what exact point “one hour after childbirth” were the interviews conducted? What happened in situation where the health workers had to attend to another delivery or perform other roles? How long did the interviews takes? Where did the interviews take place (note: the study found that visitors and relatives were a barrier to EIBF)?
--	---

	 • Page 5: Line 10-13: How many mother and healthcare providers consented and how many declined? How many had live births and stillbirths? • How did the authors mitigate against Hawthorne effects and observer bias? • Page 5. Line 24-25: “....to offer help during a life threatening situation...” How many life-threatening situations were encountered? • Page 5. Line 48: “...fatigue of the mother because of labor.” How could this have affected the findings of the study? Also, how did you deal with the problem of relatives and visitors (the study highlights this as a problem in the PHCs)? • Page 6: Line 22: Write PPI in full since it has only been used once. • Page 8: line 27-39: The information included here is a repetition of information already in table 2. Revise the section to only highlight the key findings and the reader would be able to get other findings in the table. FINDINGS  • I suggest that the authors consider reporting findings of qualitative and quantitative arms of the study separately. This will help the reader follow through their findings. Discussion section will then integrate the two study arms. • Table 1. Revise to highlight both the frequency and percentages. Report the median together with the IQR or range. • Page 11. Line 50: The verbatim is unclear and may be interpreted differently. Find a clear verbatim to use. • Page 12. Line 13-15: The sentence is unsupported and hence unclear. • How many deliveries complicated? What happened when they complicated? How many mothers were referred to other hospital for care? Were there multiple births among the observed mothers? • Table 1. Others (specify) is included in the attendant health worker during labor and delivery. Who are the others? • Parity – The use of nulliparous and multiparous should be revised. What does multiparous refer to – 2, 3, 4 or 5 deliveries? • Table 2. The reporting of the p-value should be done correctly as per the journal guidelines. • After what time was breastfeeding initiated? How many minutes after delivery? Conclusion  • Page 14. Line 19-22: “There is a 40% chance that babies born in public PHCs...” The authors should revise this statement to reflect their study findings. Also, it is important to highlight the positive rather than the focus on what is not being done i.e. 60% of the babies are breastfed within 1-hour of delivery. • Page 14. Line 29-35: “The cadre of birth attendants....” This particular conclusion is unsupported by the study. The study does not assess the association between the birth attendants cadre and EIBF. Also, most of the birth attendants in this study would not be considered “Skilled birth attendants” according to WHO definition. The author should revise or omit that conclusion. General comments  • Avoid the use of contraction e.g. wasn't, they'll etc • Use gender-neutral language e.g. instead of manpower use human resource • The paper requires English editing to address some grammatical errors • The role of the donors of SFH is not clearly stated. Being an evaluation, donor sometimes play a role in the design of evaluation study.
--	---

REVIEWER	Alyssa Sharkey UNICEF New York, USA
REVIEW RETURNED	27-Sep-2019

GENERAL COMMENTS	I think this is an interesting study and I appreciate the mixed methods used. I think the authors just need to clarify some issues within the text and then it will be a nice paper. The reviewer provided a marked copy with additional comments. Please contact the publisher for full details.
---

REVIEWER	Susan M. Walsh University of Illinois at Chicago, College of Nurs
REVIEW RETURNED	02-Oct-2019

GENERAL COMMENTS	Page 3/23, Lines 24-27 do not seem to make sense. Possibly missing words???: While international recommendations for improving breastfeeding practices in health facilities exist, the problems faced by health facilities around the practice context specific. Page 8/23, Lines 9-11 has an incomplete sentence or is a repeated fragment of the sentence before. This needs to be corrected. Page 9, Line 32 Capitalize the word 'Initiating' Page 14 Lines 12-16 We find that health education received by mothers during ANC and the post-delivery period improves the practice of early initiation of breastfeeding in public PHCs, in our study setting (suggest placing "in our study" at the start of the sentence. Page 14 Lines 31-35. The birth attendants must be trained on effective rooming-in to further improve early breastfeeding of newborns in these public PHCs, however. (either put the word "however" at the start of the sentence or else drop the word)
--

VERSION 1 – AUTHOR RESPONSE

Reviewer(s)' Comments to Author:

Reviewer: 1

Reviewer Name: Samwel Maina Gatimu

Institution and Country: Aga Khan University, Kenya

Please state any competing interests or state 'None declared': None declared

Please leave your comments for the authors below

Abstract

The authors should revise the introduction and remove the last part "...to influence health investment decisions in the region." The findings may not necessarily influence investment but could influence practice by mothers and health workers. We have revised the statement accordingly. Thanks for the suggestion.

Results: The quantitative findings of the EIBF rate during the qualitative arm of the study should also be reported. Though small, there is a 2 percentage point increase in the EIBF between the two study arms.

Use of the term “Lack of privacy...” What do the authors consider as privacy? Are delivering women accorded any privacy? Thanks for the point made. We have revised the result section of the abstract, taking your suggestions into consideration. Where ‘privacy’ is needed for instance, has been specified.

Strength and limitations

Page 2, line 33-34: “...the findings do not generalize...” to “...the findings may not...” Thanks. Changed

The last two limitations on generalization could be combined Thanks for the suggestion. We have combined them.

Introduction

- Page 3 line 27-28: Revise “...the practice context-specific” to “... the practice are context-specific” Thanks. Corrected.
- Page 3-4 line 56 and 3: “Recall bias and lack of disaggregation between..... Revise the statement to reflect the correct the available evidence on reliability and validity of maternal recall of breastfeeding. For example, from studies like <https://academic.oup.com/nutritionreviews/article-abstract/63/4/103/1940812?redirectedFrom=fulltext>. Also, highlight clearly what is limited in the utilization of the study findings from DHS. Thanks for the suggestion. We have done this.
- Page 4 Line 9-10: “the literature around the practice in PHCs in Africa is lacking.” Revise the statement to be clearer and more specific. Practice on what? Thanks. We have now clarified the statement. We meant the literature on the issues that influence early initiation of breastfeeding in PHCs is lacking.

Method

- Clearly, state the actual mixed-method design used in the study Thanks. We have stated this. The explanatory mixed methods approach is what we used.
- There exists an almost one-year difference between the quantitative and qualitative arms of the study. The author should clearly discuss why the time difference and how it affects the results or interpretation of the results of their study Thanks for the suggestion. We have explained the reason for the time difference. Also, we assume the time difference is not enough to affect the ‘practice’ of breastfeeding newborns in the study setting. .We have updated the manuscript accordingly.
- Page 4: Line 42-44: Revise the statement “The picture of maternal and newborn care in the state is comparable to other high mortality settings in sub-Saharan Africa”. The statement is too broad and the comparison with sub-Saharan Africa not right. I suggest the author consider comparing with other Nigerian states or neighbouring countries for a better context. Also, be specific on specific maternal and newborn care indicators. We have revised the statement. It is now more detailed
- What is a priority PHC? Provide a detailed description of what priority PHCs are? An explanation of what a priority PHC is has been included in the manuscript.
- What is the average delivery per day in Gombe PHCs and what were the average deliveries per day in those 10 PHCs? This information will help the reader appreciate the purposive sampling of the PHCs as well as the choice of the 3 facilities for qualitative arm of the study. Also, it is good practice to state which facilities are being referred to. This help in replication studies as well as follow-up by interested stakeholders. What was the staffing like in these facilities? Thanks for the suggestion. We however believe this information will reduce the anonymity of the selected PHCs. We decided not to include the figures.
- The author should consider providing information that would give the reader a better understanding of management of labour and childbirth in the selected PHCs. We observed all deliveries that occurred in the PHCs throughout the study period. This gives a sense of the magnitude of the deliveries in the health facilities. A reading of some of the qualitative findings raise a lot of questions on the practices e.g. after delivery, what happens? Is the child taken to a separate room? Is there a birth companion during labour/childbirth? What role do they play if present? What are the roles of the nurses – managing pregnant women and newborns or other patients within the facility? We have expanded on the readiness of these health facilities to conduct delivery services. This throws more light into delivery practices in the selected PHCs. We have also expanded on who took deliveries and

- proportion of those assisted by other health workers in the analysis section.
- o How many health care providers assisted in each delivery? This information has been included in the manuscript.
 - o There is need to provide clear conceptual or operational definition of the attendant health worker during labor and delivery e.g. who is a JCHEW, CHEW, Hospital assistant, Environmental health assistant/technician/officer. In addition, provide information about their training and scope of practice . e.g. Are they licensed to performed deliveries? What do the Nigeria and Gombe maternal and newborn policies say about them? Thanks for the suggestion. We have indicated in the manuscript that the categories mentioned above are not skilled birth attendants
 - o Was monitoring of laboring mother done by the same health attendant? What are the documentation practices? Our observation was from the second stage of labor. The attending health worker monitored the labor from that point.
 - The qualitative data analysis process should be discussed further:
 - o Page 6 Line 10: Who are two member of the team that performed the analysis? We have indicated this in the authorship contribution.
 - o What informed the choice of grounded theory? Was it the best choice of qualitative approach? Thanks for the question. We used thematic analysis. We have corrected this in the manuscript.
 - o In what language were the interviews conducted? In English and Hausa. We have indicated this in the manuscript.
 - o How was grounded theory used? Grounded theory was written in error.
 - o At what point was analysis of data performed? Same as above
 - o How did the authors ascertain theoretical saturation? Same as above
 - o Include either in text or as appendix – the coding process – how the current themes were arrived at. Coding process elaborated in the manuscript.
 - Page 6. Line 16: Delete the line “To analyze the respondents...” deleted.
 - How was consent obtained without informing the healthcare workers and mother what was being observed? The author should consider including sample information sheet and consent forms used as supplementary materials. This has been included.
 - It would be informative for the authors to include the data collection tools as supplementary materials This will be shared with reasonable request.
 - At what exact point “one hour after childbirth” were the interviews conducted? What happened in situation where the health workers had to attend to another delivery or perform other roles? How long did the interviews takes? Where did the interviews take place (note: the study found that visitors and relatives were a barrier to EIBF)? Thanks for raising these questions. I
 - Page 5: Line 10-13: How many mother and healthcare providers consented and how many declined? How many had live births and stillbirths? This information has been updated in the manuscript.
 - How did the authors mitigate against Hawthorne effects and observer bias? We did not alert the health workers or mothers of the particular aspect of newborn care that we were assessing while consenting was being sort.
 - Page 5. Line 24-25: “.....to offer help during a life threatening situation...” How many life-threatening situation were encountered? None. This was only anticipated.
 - Page 5. Line 48: “...fatigue of the mother because of labor.” How could this have affected the findings of the study? Also, how did you deal with the problem of relatives and visitors (the study highlights this as a problem in the PHCs)? Fatigue/tiredness is one of the things we observed to be a barrier to breastfeeding newborns within the hour. We excused visitors/relatives during the interviews. This has been explained in the manuscript.
 - Page 6: Line 22: Write PPI in full since it has only been used once. Written out.
 - Page 8: line 27-39: The information included here is a repetition of information already in table 2. Revise the section to only highlight the key findings and the reader would be able to get other findings in the table. Thanks. The section has been revised to reflect key findings.

FINDINGS

- I suggest that the authors consider reporting findings of qualitative and quantitative arms of the study separately. This will help the reader follow through their findings. Discussion section will then integrate the two study arms. Thanks for the suggestion. Because we used an explanatory mixed methods approach, we planned to integrate the results of the two arms to improve understanding.
- Table 1. Revise to highlight both the frequency and percentages. Report the median together with the IQR or range. Thanks. We have now done this.
- Page 11. Line 50: The verbatim is unclear and may be interpreted differently. Find a clear verbatim to use. Explanatory words in brackets have been inserted into the verbatim to make it clearer.
- Page 12. Line 13-15: The sentence is unsupported and hence unclear. Thanks for the suggestion. Not sure we understand why it is unclear.
- How many deliveries complicated? What happened when they complicated? How many mothers were referred to other hospital for care? Were there multiple births among the observed mothers? Thanks for the suggestion. We have updated the manuscript to indicate the proportion of multiple births that occurred. We also reported that complications like blood loss did not affect the initiation of breastfeeding.
- Table 1. Others (specify) is included in the attendant health worker during labor and delivery. Who are the others? We have now expanded on this.
- Parity – The use of nulliparous and multiparous should be revised. What does multiparous refer to – 2, 3, 4 or 5 deliveries? This has now been revised in the manuscript.
- Table 2. The reporting of the p-value should be done correctly as per the journal guidelines. We have now done this.
- After what time was breastfeeding initiated? How many minutes after delivery? We focus the study to look at the cut-off point i.e. whether breastfeeding occurred within or after an hour. Another study is looking at the time distribution.

Conclusion

- Page 14. Line 19-22: “There is a 40% chance that babies born in public PHCs...” The authors should revise this statement to reflect their study findings. Also, it is important to highlight the positive rather than the focus on what is not being done i.e. 60% of the babies are breastfed within 1-hour of delivery. Many thanks for the suggestion. We have revised this.
- Page 14. Line 29-35: “The cadre of birth attendants...” This particular conclusion is unsupported by the study. The study does not assess the association between the birth attendants cadre and EIBF. Also, most of the birth attendants in this study would not be considered “Skilled birth attendants” according to WHO definition. The author should revise or omit that conclusion. Thanks for this. We have revised the statement.

General comments

- Avoid the use of contraction e.g. wasn't, they'll etc done.
- Use gender-neutral language e.g. instead of manpower use human resource done
- The paper requires English editing to address some grammatical errors grammatical errors corrected.
- The role of the donors of SFH is not clearly stated. Being an evaluation, donor sometime play a role in the design of evaluation study. We have further elaborated on this.

Reviewer: 2 *Further comments are provided in the attached file*

Reviewer Name: Alyssa Sharkey

Institution and Country: UNICEF New York, USA

Please state any competing interests or state 'None declared': None declared

Please leave your comments for the authors below

I think this is an interesting study and I appreciate the mixed methods used. I think the authors just need to clarify some issues within the text and then it will be a nice paper. Many thanks for the

feedback. We appreciate it.

Reviewer: 3

Reviewer Name: Susan M. Walsh

Institution and Country: University of Illinois at Chicago, College of Nurs

Please state any competing interests or state 'None declared': None declared

Please leave your comments for the authors below

Page 3/23, Lines 24-27 do not seem to make sense. Possibly missing words???: While international recommendations for improving breastfeeding practices in health facilities exist, the problems faced by health facilities around the practice context specific. Thanks for the observation. We have corrected the error.

Page 8/23, Lines 9-11 has an incomplete sentence or is a repeated fragment of the sentence before. This needs to be corrected.

Page 9, Line 32 Capitalize the word 'Initiating' done

Page 14 Lines 12-16 We find that health education received by mothers during ANC done and the post-delivery period improves the practice of early initiation of breastfeeding in public PHCs, in our study setting (suggest placing "in our study" at the start of the sentence. done

Page 14 Lines 31-35. The birth attendants must be trained on effective rooming-in to further improve early breastfeeding of newborns in these public PHCs, however. (either put the word "however" at the start of the sentence or else drop the word) done.

VERSION 2 – REVIEW

REVIEWER	Samwel Maina Gatimu Aga Khan University, Kenya
REVIEW RETURNED	22-Jan-2020

GENERAL COMMENTS	The authors addressed most of the comments and provided reasonable explanations for those not fully addressed. The authors should address the following minor issues: • The statement on Page 5 Line 41-44 is not supported by the study findings. The study found that there was a difference in the EIBF rate between the quantitative (61%) and qualitative (63%) arms. The 2-percentage points might seem small but the inclusion of confidence intervals for each rate could help highlight whether the difference is significant, despite the differences in the sample sizes.• Table 1. Capitalise the appropriate words in table 1 and 3• Page 4. Line 3: change "will" to "can"• Revise page 10 line 38-40 to make it clearer. E.g. "They attributed their knowledge to on-the-job training sessions on newborn care had with visiting clinical-mentors." to "They attributed their knowledge to on-the-job training sessions on newborn care with visiting clinical-mentors." OR "They attributed their knowledge to on-the-job training sessions on newborn care they had with visiting clinical-mentors."• Table 3. All predictors variables have not been included. p-value of 1 should be revised to >0.9.• Consider revising the sentence (Page 15, Line 9-15) to "It is also higher than estimates amongst mothers who had spontaneous vaginal deliveries (SVD) in some secondary health facilities; and
---

	lower than estimates amongst mothers who had SVD in some tertiary health facilities, even in Nigeria.” to make it clearer.  • The relative risks on page 12 lines 39 and 43: The two RR are not included in Table 3. Consider including them in the table for consistency especially noting that one is statistically significant when most predictor variables are not statistically significant.
--	--

VERSION 2 – AUTHOR RESPONSE

- The statement on Page 5 Line 41-44 is not supported by the study findings. The study found that there was a difference in the EIBF rate between the quantitative (61%) and qualitative (63%) arms. The 2-percentage points might seem small but the inclusion of confidence intervals for each rate could help highlight whether the difference is significant, despite the differences in the sample sizes.:

Thank you for noting this. We have resolved the issue by including the confidence intervals for the estimates on page 7 line 22 and page 8 line 6 of the “main document”.

- Table 1. Capitalise the appropriate words in table 1 and 3:

Thank you for the suggestion. We have resolved the issue on page 7 line 29 and page 10 line 10 of the clean copy.

- Page 4. Line 3: change “will” to “can”:

We appreciate the suggestion. We have updated the manuscript accordingly on page 3 line 1 of the “main document”.

- Revise page 10 line 38-40 to make it clearer. E.g. “They attributed their knowledge to on-the-job training sessions on newborn care had with visiting clinical-mentors.” to “They attributed their knowledge to on-the-job training sessions on newborn care with visiting clinical-mentors.” OR “They attributed their knowledge to on-the-job training sessions on newborn care they had with visiting clinical-mentors.”:

Thank you for the suggestion. We have updated the sentence as advised on page 9 line 13-15 of the “main document”

- Table 3. All predictors variables have not been included. p-value of 1 should be revised to >0.9:

We have now included all predictor variables in Table 3. Thank you for noting it.

- Consider revising the sentence (Page 15, Line 9-15) to “It is also higher than estimates amongst mothers who had spontaneous vaginal deliveries (SVD) in some secondary health facilities; and lower

than estimates amongst mothers who had SVD in some tertiary health facilities, even in Nigeria.” to make it clearer.:

Thank you for the suggestion. The revised sentence is on page 8 line 8-11 of the main document

- The relative risks on page 12 lines 39 and 43: The two RR are not included in Table 3. Consider including them in the table for consistency especially noting that one is statistically significant when most predictor variables are not statistically significant.:

Thank you for noting this. We have revised Table 3 to include the corresponding variables and RRs in the “main document”

VERSION 3 – REVIEW

REVIEWER	Samwel Maina Gatimu Aga Khan University, Kenya
REVIEW RETURNED	04-Mar-2020
GENERAL COMMENTS	The authors have satisfactorily addressed all the previous comments.

VERSION 3 – AUTHOR RESPONSE

Reviewer(s)' Comments to Author:

Reviewer: 4

Reviewer Name: Samwel Maina Gatimu

Institution and Country: Aga Khan University, Kenya

Please state any competing interests or state 'None declared': None declared

Response:

Thank you for your suggestion. We have reviewed the statement to read 'None Declared'. You will find the change on Page17 line13. Thank you.